# Mechanical Properties and Formaldehyde Release of Particleboard Made with Lignin-Based Adhesives

**Gabriela Balea Paul, Maria Cristina Timar** [ID]**, Octavia Zeleniuc, Aurel Lunguleasa** [ID] **and Camelia Coșereanu \*** [ID]

Faculty of Furniture Design and Wood Engineering, Transilvania University of Brasov, B-dul Eroilor nr. 29, 500036 Brasov, Romania; gabriela.balea@unitbv.ro (G.B.P.); cristinatimar@unitbv.ro (M.C.T.); zoctavia@unitbv.ro (O.Z.); lunga@unitbv.ro (A.L.)
\* Correspondence: cboieriu@unitbv.ro

**Abstract:** The aim of this research was to evaluate the potential of magnesium lignosulfonate as adhesive in particleboard manufacturing. Diphenylmethane diisocyanate (PMDI) between 1% and 3% and glucose (1% of the lignosulfonate content) were added as potential cross-linkers in the adhesive formulations. Mixed beech and spruce wood, 30% beech wood and 70% spruce wood, were employed for the configuration of the panel structure. The density, mechanical properties and formaldehyde emission of single-layer particleboard were investigated. Spectroscopic analysis (FTIR) revealed structural changes brought by oxidation that may indicate depolymerization by the splitting of C-O-C bonds and formation of carbonyl groups. Mechanical properties were improved, and the highest average values were recorded for panels having as adhesives oxidized lignin with cross-linkers as follow: 15 N/mm$^2$ (MOR), 3320 N/mm$^2$ (MOE) and 0.48 N/mm$^2$ (IB). The density profile presented higher values for faces in case of oxidized lignin panels. Changes were observed for oxidized lignin with cross-linker panels wherein the core had higher values. The results showed that the panels manufactured with adhesives composed of oxidized lignosulfonate (20% of the dried wood particles weight) and the addition of PMDI and glucose in various percentages have a positive influence on their formaldehyde release and mechanical properties requested by EN 312 (2004) standard.

**Keywords:** wood particleboard; magnesium lignosulfonate; mechanical properties; formaldehyde emission; FTIR; bio-based adhesives; wood-based composites

## 1. Introduction

Formaldehyde emission represents a key issue for the wood-based composite industry. Formaldehyde issues are related to urea-formaldehyde resin (UF) as a dominating bonding adhesive used in the production of wood-based panels [1,2]. Melamine-formaldehyde (MF), melamine-urea-formaldehyde (MUF) and phenol-formaldehyde (PF) adhesives are less dangerous in terms of their subsequent formaldehyde emission, but they have not convinced industrial producers, due to their higher price or lower reactivity [3,4].

Researchers use various methods to modify the traditional urea-formaldehyde (UF), formulations [5,6] in order to meet regulations concerning formaldehyde release. Exposure to formaldehyde is possible during the adhesive-mixing, mat-forming and hot-pressing operations. Formaldehyde exposure can be harmful to human health. At low levels, formaldehyde can cause eye, nose and throat irritation, and skin rashes, shortness of breath, wheezing and changes in lung function at higher levels of exposure [7]. The International Agency for Research on Cancer (IARC) has reclassified formaldehyde into Group 1-"carcinogenic to humans" [8].

In Europe, the European Chemicals Agency (ECHA) reclassified formaldehyde in category 1B "presumed human carcinogen" and "germ cell mutagen category 2 (acute toxicity)" [9]. With growing interest in indoor air quality, efforts have been made to reduce

exposure limits to formaldehyde both in workplaces and in living spaces. There is legislation in the European Union, USA, China, and Japan limiting the levels of formaldehyde emission (FE) from wood and wood-based products.

The limits of formaldehyde emissions set by CARB (California Air Resources Board) are the following: 0.05 ppm for hardwood plywood, 0.09 ppm for particleboard, 0.11 ppm for MDF (medium-density fibreboard) and 0.13 ppm for thin MDF (up to 8 mm thick) [10]. Such regulations established the allowable limits of formaldehyde emissions at a level 10 to 20 times lower than that that existed 30 years ago [2], and, in a few years, the limits could be lowered, forcing producers to adopt alternative methods. Among these, several methods can be mentioned: binderless particleboard manufacturing [11], the use of formaldehyde scavengers [4,12–15], the surface treatment of wood composites [16,17] and the use of bio-adhesives prepared from natural raw materials, such as lignin [18–20], starch [21,22], soy or tannin [23]. The studies were especially focused on three biopolymers: lignin, starch and plant proteins, but lignin seems to be the most used one in the experimental research and industrial trials to manufacture particleboards.

Lignin-based adhesives can be classified into two groups: formaldehyde-free lignin-based adhesives and lignin phenol formaldehyde adhesives, when lignin is used as a partial replacement of phenol [24,25]. For the first group of adhesives, mixtures between kraft lignin and polyethylenimine [26], lignin with added glyoxal [27], or oxygen-plasma-treated enzymatic hydrolyzed lignin [28] were investigated. A study [29] referred to lignin-based copolymer adhesives for the production of wood based composites and found that up to 50%wt. of the phenol can be replaced by kraft lignin. Other study [30] found that up to 50% of phenol in phenol formaldehyde adhesive could be substituted by bark lignin with improved adhesive properties.

Lots of research has been done to improve the reactivity of lignin as the substitute of phenol in phenol-formaldehyde (PF) adhesive synthesis by modifying the chemical structure of lignin to increase its reactivity and reduce the long pressing time and high pressing temperature [31]. The most-used methods to fulfill this objective are [32,33] methylolation (hydroxymethylation), phenolation, demethylation, reduction, hydrolysis and oxidation [31,34–36]. The oxidation of lignin has been described as a way to weaken the lignin structure, making it more susceptible to depolymerization [36] and a good procedure to improve the properties of lignin [31]. An approach using mild reaction conditions is the base-catalyzed depolymerization of lignin and the addition of the hydrogen peroxide, considered to be an environmentally friendly oxidant [37].

The contribution of lignosulfonates to the performance of the engineered wood panels is presented in several papers. Calcium lignosulfonate [38] magnesium and sodium lignosulfonates [39], ammonium lignosulfonate [40–42] or modified ammonium lignosulfonate [43] were used as adhesives for lowering formaldehyde emission and improve the mechanical properties of the panels. A mixture of phenol formaldehyde adhesives and lignosulfonate (up to 20% of lignosulfonate in the adhesives) was shown to improve the shear strength of wood glued joints gradually [44]. The incorporation of lignosulfonates may replace between 40% and 70% of PF adhesive [33]. Research on phenol replacement by different quantities and types of lignin up to a maximum of 40wt.% resulted in the best performance for pine kraft lignin with a phenol substitution degree of 20% [45]. The wider industrial application of lignosulfonates in the composition of wood adhesives is limited by the increased hydrophilicity of finished wood-based panels, longer press times and the availability of crosslinkers [46,47]. Other results indicated that organosolv lignin was a feasible replacement for up to 30% of the phenol in particleboard-type PF adhesives [48]. Recent research works [47,49] have shown that magnesium lignosulfonate (15%wt. reported to wood) is a suitable adhesive for obtaining eco-friendly fiberboards with satisfactory mechanical and physical properties. Their remarkably low formaldehyde content (1.1 mg/100 g, super E0 grade) made these panels suitable for interior design. Pressure from environmental and health regulators led to formaldehyde emission requirements becoming more stringent. Thus, super-E0 ($\leq$1.5 mg/100 g acc. to Perforator method EN

ISO 12460-5. or $\leq$0.3 mg/L acc. to Desiccator method JIS A 1460) and E0 ($\leq$2.5 mg/100 g, or $\leq$0.5 mg/L) classes of panels are promoted [50].

Other investigations were conducted with the aim to study lignosulfonates with the addition of cross-linkers. Studies showed improvement of particleboards performance by using ammonium lignosulfonate with polymeric diphenylmethane diisocyanate (PMDI) and furfuryl alcohol as cross-linkers. PMDI is a promising cross-linker that works for most bio-based adhesives [39,51,52], due to the reactive isocyanate's groups. Sugars, as cross-linkers, bear both primary and secondary alcohol groups, which theoretically could be used for crosslinking reactions. Other crosslinkers are mentioned in the literature: alternative aldehydes (glyoxal, glutaraldehyde), polyacids (citric acid, maleic anhydride) [52].

The present paper aims to evaluate the mechanical properties and formaldehyde emission of particleboards made with magnesium lignosulfonate-based adhesives. The influence of cross-linkers such as PMDI and glucose on the formaldehyde release and mechanical performance of the panels has been also investigated.

## 2. Materials and Methods

### 2.1. Materials

2.1.1. Wood Particles

Raw materials were supplied by Kastamonu particleboard manufacturer (Romania), and contained mixed beech (30%) and spruce (70%) wood particles at a moisture content of around 10%, with a bark percentage of 5% of the total amount [53]. The granulometric analysis of the particles was carried out using the horizontal vibrating sieves (VEB Metall-weberei Neustadt Orla, Germany). The particles remained in the 4 mm $\times$ 4 mm, 3.15 mm $\times$ 3.15 mm, and 2 mm $\times$ 2 mm sieves were classified as coarse particles, in a representation rate of 13.8%, 5.7%, and 80.5% respectively. The fine particles were retained in the 1.25 mm $\times$ 1.25 mm, 1 mm $\times$ 1 mm and 0.8 mm $\times$ 0.8 mm sieves in the following percentages: 57.9%, 22.9% and 19.2% for the smallest mesh sieve. The sizes of the particles are presented in Table 1.

**Table 1.** Sizes (Range of Values) of Wood Particles.

| Sieve Size mm $\times$ mm | Particles Sizes (Range of Values), in mm | | |
| --- | --- | --- | --- |
| | Length | Width | Thickness |
| 4.00 $\times$ 4.00 | 7.6–25.8 | 4.1–10.6 | 0.2–4.1 |
| 3.15 $\times$ 3.15 | 6.1–18.0 | 4.1–5.7 | 0.4–3.1 |
| 2.00 $\times$ 2.00 | 4.2–34.1 | 1.1–5.2 | 0.2–1.8 |
| 1.25 $\times$ 1.25 | 3.7–25.6 | 0.9–3.4 | 0.1–1.6 |
| 1.00 $\times$ 1.00 | 2.4–19.5 | 0.5–1.7 | 0.2–0.9 |
| 0.8 $\times$ 0.8 | 2.0–5.0 | 0.2–0.7 | 0.2–0.5 |

The slenderness ratio, fatness ratio and width factor were calculated to obtain the particle geometric characteristics [54]. The slenderness (length to thickness ratio) was of 15.3 for the fines and of 8.9 for the coarse particles. Fatness (width-to-thickness ratio) was 2.11 for the fines and 2.72 for the coarse particles, and the width factor (length to width ratio) varied from 7.7 to 3.27 for the fines and coarse particles.

2.1.2. Adhesives

A bio-adhesive based on magnesium lignosulfonate, a yellow–brown colored powder was used in the present research. This powder, coded LIGNEX MG F, was supplied by SAPPI BIOTECH GmbH (Düsseldorf, Germany). The characteristics of the magnesium lignosulfonate LIGNEX MG F are presented in Table 2.

In order to increase the lignin reactivity, the magnesium lignosulfonate was oxidized with 30% hydrogen peroxide ($H_2O_2$) in an aqueous alkaline solution. The $H_2O_2$ content was 7.5% of the lignosulfonate weight, and sodium hydroxide (NaOH) was used to increase the

pH to 9. The characteristics of the oxidized magnesium lignosulfonate (OML) are presented in Table 3.

**Table 2.** Characteristics of the Magnesium Lignosulfonate LIGNEX MG F.

| Characteristic | Description/Value |
|---|---|
| pH Value | $5.5 \pm 1\%$ |
| Dry matter mass | $93 \pm 2\%$ |
| Insolubility in water | 1% max |
| Magnesium | $6 \pm 1\%$ min |
| Bulk density | 400 kg/m$^3$ |
| Moisture content | 7% max |

**Table 3.** Characteristics of the oxidized magnesium lignosulfonate (OML).

| Material | Values |
|---|---|
| Solid content | 58.4% |
| pH | 9 |
| Viscosity (flow time through the viscosmetric STAS cup, $\Phi$ 6 mm, 20 $^\circ$C) | 16 s |
| Adhesive reactivity at 160 $^\circ$C | 3 min 15 s |

The reactivity of magnesium lignosulfonate, representing the curing time at 160 $^\circ$C determined similarly as for the PF adhesives, was reduced from 5 min 20 s to 3 min 15 s, by 40%, through the oxidation process.

Two types of cross-linkers, namely PMDI (100% solid, provided by Kastamonu S.A. Romania) (from 1% to 3% of the weight of the dried wood particles) and glucose (1% of the magnesium lignosulfonate content) were also added to the recipes in order to improve the mechanical properties of particleboard and to keep formaldehyde emission at lower level. First, the adhesive recipes were modified by adding PMDI amounts in the range of 1%, 2% and 3% as wt. reported to the weight of the dried wood particles. The second modification of the adhesive was done by adding glucose as a cross-linker (1%wt. reported to the lignosulfonate content) to the recipes with PMDI amounts of wt.1% and wt.2%, respectively (reported to the weight of the dried wood particles).

The magnesium lignosulfonate (LIGNEX MG F) content in all particleboards' recipes was 20% of the weight of the dried wood particles.

### 2.2. Particleboard Manufacturing

The particles (65% coarse and 35% fine particles) were mixed together to obtain a single-layered structure. The established target density was 650 kg/m$^3$, comparable to that of the urea-formaldehyde particleboards manufactured at Kastamonu SA Romania.

The adhesive and the wood particles were mixed mechanically (with a mixer with pallets made by self-endowment) for 10 min. The mixture was then placed in the forming frame and prepressed manually with a metal plate, then the frame was removed and the mat was hot-pressed in the laboratory press (Metrom, Brasov, Romania) at 180 $^\circ$C for 16 min under a pressure of 2.5 N/mm$^2$. After being removed from the press, the panels with nominal dimensions of 400 mm $\times$ 400 mm $\times$ 16 mm (thickness) were conditioned at a temperature of 20 $^\circ$C and a relative humidity of 65%, for 7 days. The experimental panels were cut into samples for mechanical and formaldehyde emission testing. Three replicates for each type of panels were manufactured.

The oxidized magnesium lignosulfonate adhesive (solid content of 58.4%) was used in the proportion of 20% of the wood particles' dry mass. The other adhesives, as shown in Table 4, were obtained from oxidized magnesium lignosulfonate and the gradually addition of PMDI in proportion of 1%, 2% and 3%, respectively (based on the particles' dry mass). Glucose was added at wt.1% of the magnesium lignosulfonate weight to the recipes with PMDI proportions of 1% and 2%, respectively.

**Table 4.** Composition of the adhesives and the panels' identification codes.

| Adhesive Composition for the Manufactured Panels | Panels' Codes |
| --- | --- |
| Magnesium lignosulfonate powder | L20 |
| Oxidized magnesium lignosulfonate | LO 20 |
| Oxidized magnesium lignosulfonate + 1% PMDI | LO 20 P1 |
| Oxidized magnesium lignosulfonate + 2% PMDI | LO 20 P2 |
| Oxidized magnesium lignosulfonate + 3% PMDI | LO 20 P3 |
| Oxidized magnesium lignosulfonate + 1% PMDI + glucose | LO 20 P1G |
| Oxidized magnesium lignosulfonate + 2% PMDI + glucose | LO 20 P2G |

*2.3. FTIR*

Fourier transform infrared spectroscopy (FTIR) was performed for the magnesium lignosulfonate powder (LIGNEX MG F), and the adhesives were prepared according to the compositions presented in Table 4. For all the adhesive recipes, cross-linked (cured) adhesive samples were investigated by FTIR. Crosslinking (curing) of the adhesives was performed at 160 °C for 15 min. This process was achieved in a laboratory oven (Binder ED 115, Tuttlingen, Germany), for the unmodified lignosulfonate, while cured adhesive samples were extracted under microscope from the particleboards glued with modified magnesium lignosulfonate. For magnesium lignosulfonate, sample preparation was performed by mixing the powder (10 parts) with water (1 part): the mixture was applied as a film on microscope slides and allowed to dry at room temperature or cured at 160 °C/15 min in an oven.

FTIR spectra were recorded employing an ALPHA Bruker spectrometer (produced by Bruker Optik GmbH, Germany) equipped with an ATR (attenuated total reflection), module in the range 4000–400 cm$^{-1}$ at a resolution of 4 cm$^{-1}$ and 24 scans/spectrum. Three spectra were recorded for each type of samples; these were further processed for baseline correction and smoothing, and average spectra were computed. The average spectra were further normalized (Max-Min normalization) and compared in order to highlight any chemical changes due to curing at 160 °C temperature of magnesium lignosulfonate or its modification by oxidation and addition of the two tested cross-linkers. The assignment of characteristic absorption bands was based on references in the literature.

*2.4. Mechanical Testing of the Particleboard*

The number and sizes of the specimens used for mechanical tests were according to the European standards: EN 310:1993 for bending strength (MOR) and modulus of elasticity (MOE) [55] and EN 319:1993 for internal bond (IB) perpendicular to the plane of the panel [56]. The results were compared to the limits imposed by the European standard for particleboard specifications EN 312 [57].

*2.5. Microscopic Investigation and Density Profile along the Thickness*

The microscopic investigation was conducted with the stereo-microscope NIKON SMZ 18-LOT2 (Nikon Corporation, Tokyo, Japan), with 30× and 180× magnification. The microscopic investigation was performed on the edges of the samples prepared for IB testing in order to analyze the interface between wood particles and adhesive, observing in the same time, the structure defects that could affect the mechanical properties of the panels. Complementary to the microscopic analysis, the density profile along the thickness will give information about the behavior of the particleboard panel to IB, MOE and MOR testing.

The density profiles along the thickness were analyzed using the X-ray density profile analyzer DPX300 (IMAL, San Damaso, Italy). The vertical density profiles were measured on five specimens for each panel type. The specimens had sizes of 50 mm × 50 mm × 16 mm, and the density profile was measured on a thickness of 16 mm.

*2.6. Free Formaldehyde Emission*

Formaldehyde emission was determined using the gas analysis method [58] and Timber Test equipment (New Plymouth, New Zealand). The evaluation of formaldehyde emission was performed after the panels conditioning to constant mass. Moisture content and densities of the panels were determined according to the requirements of the European standards [59,60]. The gas analysis is a derived test that determines formaldehyde emission under accelerated conditions at a temperature of 60 °C and within a period of 4 h. The test samples with dimensions of 400 mm × 50 mm × 16 mm and edges sealed with aluminum tape were placed in a closed chamber where the conditions (temperature, relative humidity less than 3% and air flow of $(60 \pm 3)$ L/h and pressure between 1000 Pa and 1200 Pa) were controlled during the test. After 4 h of testing, the concentration of formaldehyde in water was photometrical determined. The formaldehyde emission (in $mg/m^2 \cdot h$) was calculated based on this concentration, the sampling time and the exposed area of the sample. The tests were performed on two replicates, using two different samples for each type of panel, and the average value obtained should be less than 3.5 $mg/m^2$ to be classified into the E1 emission class, according to the European standard [61].

*2.7. Statistical Methods*

Minitab 18 statistical software (Coventry, UK) was used for the interpretation of the differences between the tested lignin-based adhesives, related to mechanical properties. A confidence interval of 95% was used for the statistical analysis, with the acceptance of an alpha type error of 0.05. The Anderson–Darling and *p*-value parameters analyzed the normality of the distribution of the results and whether it has statistically significant differences.

## 3. Results and Discussions

*3.1. FTIR*

The spectroscopic analysis for magnesium lignosulfonate (LIGNEX MG F) is presented in Figure 1. This analysis was done for three variants of the lignosulfonate, as follows: non-modified—as a powder in the initial state, mixed with water in the ratio 10:1 and air-dried, and as a crosslinked adhesive at 160° for 15 min resulting from the mixture of magnesium lignosulfonate with water in the ratio 10:1.

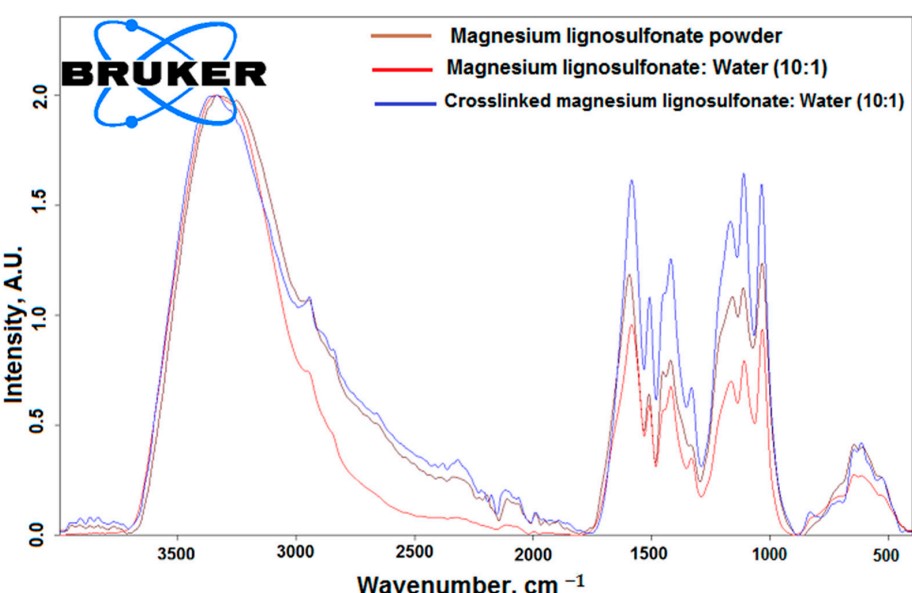

**Figure 1.** FTIR analysis for (LIGNEX MG F) powder and mixed LIGNEX MG F—water (air-dried and crosslinked).

The recorded FTIR spectra in Figure 1 present the characteristic absorption bands of the OH vibration at 3334 $cm^{-1}$ and the C-H vibration in the methyl and methylene groups at 2945 $cm^{-1}$ (small or shoulder). The most numerous absorption bands were observed in the fingerprint region between 1600 $cm^{-1}$ and 600 $cm^{-1}$. A pronounced band at 1595 $cm^{-1}$ may be assigned to aromatic skeletal vibration (originating from the aromatic C=C stretching). The band at 1512 $cm^{-1}$ was assigned to skeletal vibration of lignin, while the band at 1453 $cm^{-1}$ may be assigned to C-H deformation in aliphatic $CH_3$ and $CH_2$ groups, as observed by other researchers at 1465 $cm^{-1}$ [62]. A weak band at 1421 $cm^{-1}$ can be assigned to the C-H deformation in lignin and a band at 1332 $cm^{-1}$, which appears small as a shoulder, was found to belong to syringyl ring breathing [63], aliphatic O-H bending [64] or C-H in plane deformation of cellulose rings [65]. The band at 1160 $cm^{-1}$ can be assigned to asymmetric $–SO_2-$ vibration, with reference to 1169 $cm^{-1}$ corresponding to asymmetric (at 1169 $cm^{-1}$) and symmetric (at 1142 $cm^{-1}$) $-SO_2-$ vibrations [66], whereas a double band at 1143 $cm^{-1}$ and 1169 $cm^{-1}$ was assigned to binding vibrations in the aliphatic C-OH [63]. The 1114 $cm^{-1}$ band is assigned to C-O-C vibration and 1034 $cm^{-1}$ is assigned to C-O vibration of the phenolic OH [67] and/or $-SO_3H$ group [62], or Ar-O-C [35]. The peak at 646 $cm^{-1}$ is assigned to S-O vibration in lignosulfonates.

The comparative spectra of magnesium lignosulfonate powder, the air-dried adhesive prepared from magnesium lignosulfonate powder and water (10 to 1) and the crosslinked adhesive at 160 °C for 15 min (Figure 1), are similar in terms of absorption bands, indicating a similar chemical structure. There are some differences in the relative intensities of the absorption bands in the range 1600–1000 $cm^{-1}$ compared to the –OH absorption at 3400 $cm^{-1}$ (with quasi-constant intensity due to the min-max normalization of the spectra). As any water present in the analyzed samples contributes to the absorption at 3400 $cm^{-1}$, these differences might be related to different water content in the three types of samples. For the adhesive prepared and crosslinked, the absorption at 1332 $cm^{-1}$ is better highlighted. The similar spectra indicate a physical mechanism of bonding, based mainly on the thermoplastic character of lignin.

Structural changes (Figure 2) brought by oxidation (LO 20 compared to L20) are, as follows: the appearance of a shoulder at 1728 $cm^{-1}$ assignable to unconjugated carbonyl groups; the shift of the aromatic skeletal vibration from 1590 $cm^{-1}$ to 1601 $cm^{-1}$; a shoulder at 1368 $cm^{-1}$ specific to C-H deformation; a small shoulder at 1212 $cm^{-1}$, indicating C-O vibration in guaiacyl ring; a shift of absorption from 1160 $cm^{-1}$ (assigned to –SO2- and C-OH) to 1150 $cm^{-1}$, and its significant decrease; a shift of C-O-C vibration from 1114 $cm^{-1}$ to 1106 $cm^{-1}$, and its significant decrease. These may indicate depolymerization through the splitting of C-O-C bonds and oxidation leading to carbonyl groups, which is in accordance with the expected effects of mild oxidation [37].

Structural changes due to the addition of PMDI to oxidized magnesium lignosulfonate (LO 20 P1, LO 20 P3), as seen in Figure 2, have shown increased shoulder at 1730 $cm^{-1}$, indicating carbonyl CO groups in urethane structure; an increase of absorption at 1216 $cm^{-1}$ (LO 20 P1), which is characteristic for transformation of –NCO groups into urethane structures; an increase of 1105 $cm^{-1}$, indicating C-O-C groups, (more visible for LO 20 P3).

Structural changes due to the addition of glucose as a cross-linker to oxidized lignin (LO 20 P1G, LO 20 P2G) were noticed (Figure 2), such as: an increase of the 1422 $cm^{-1}$ absorption band and an increase of the 1152 $cm^{-1}$ band, which cumulates also the absorption at 1220 $cm^{-1}$. For LO 20 P1G, a shoulder at the 1760 $cm^{-1}$ absorption band, a small absorption peak at 1700 $cm^{-1}$ (carbonyl group in urethane structure), a decrease of lignin skeletal vibration at 1515 $cm^{-1}$, and increased absorption at 1216 $cm^{-1}$, which is characteristic to urethane structures, were noticed. All these might implicate glucose and lignin's roles in crosslinking with PMDI.

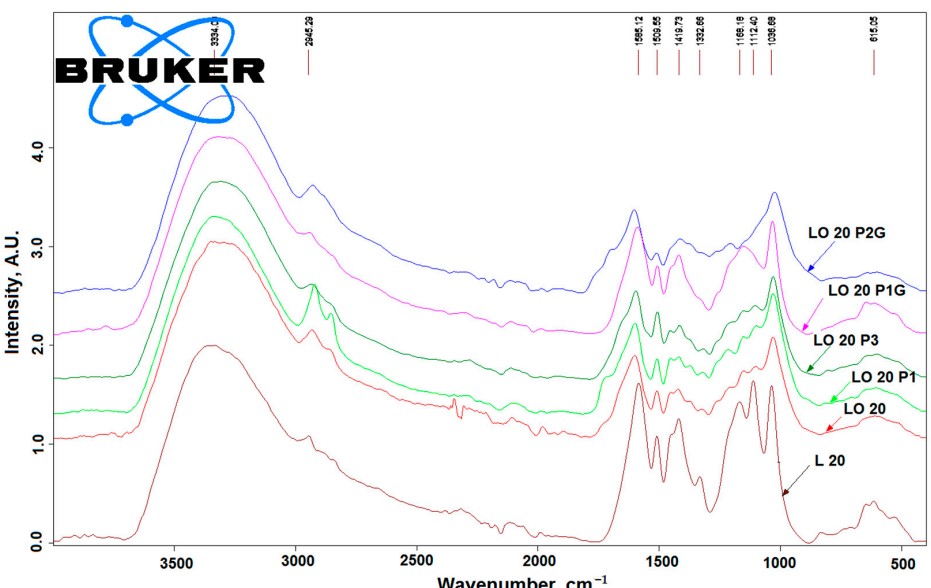

**Figure 2.** FTIR analysis for oxidized LIGNEX MG F and cross-linked adhesives with PMDI and glucose.

### 3.2. Mechanical Properties

The results of the mechanical testing of the experimental panels are presented in Table 5, where the values in the parentheses represent the standard deviations. As a general remark, the mechanical properties of the particleboard panels are improved for the modified magnesium lignosulfonate thought oxidation and further, by adding cross-linkers, such as PMDI and glucose.

**Table 5.** Experimental results on the mechanical properties of the tested panels.

| Panel Type | MOE [1] (N/mm$^2$) | MOR [1] (N/mm$^2$) | IB [1] (N/mm$^2$) |
|:---:|:---:|:---:|:---:|
| L 20 | 1847 (133) | 8.1 (0.94) | 0.10 (0.03) |
| LO 20 | 2783 (319) | 10.6 (0.76) | 0.12 (0.03) |
| LO 20 P1 | 2526 (185) | 11.0 (0.34) | 0.22 (0.03) |
| LO 20 P2 | 2778 (239) | 13.0 (0.47) | 0.31 (0.05) |
| LO 20 P3 | 2931(89) | 15.3 (0.73) | 0.41 (0.04) |
| LO 20 P1G | 3123 (152) | 13.3 (0.74) | 0.38 (0.06) |
| LO 20 P2G | 3320 (112) | 14.3 (0.58) | 0.48 (0.07) |

[1] Values in the parenthesis are standard deviations.

The values of MOR and MOE are comparable with results obtained for composite panels produced from waste fibers bonded with magnesium lignosulfonate [47]. The comparison between the mechanical performances of the experimental panels and the results of the modulus of elasticity (MOE), bending strength (MOR) and internal bond, (IB) in relation with the limits imposed by EN 312: 2004 [57] for the panel P2 type (designed for the indoor application, including furniture), is presented in Figure 3.

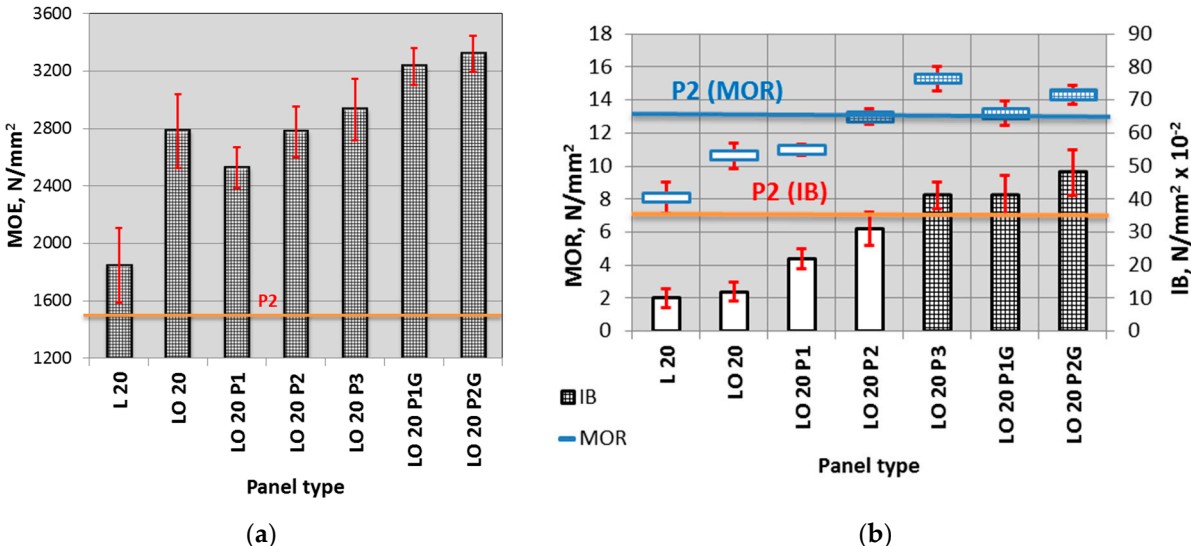

**Figure 3.** Experimental results on the mechanical properties of the particleboard made with lignin-based adhesives: (**a**) modulus of elasticity (MOE); (**b**) modulus of rupture (MOR) and internal bond (IB).

As seen in Figure 3a, MOE increased 1.8 times for the particleboard made with oxidized lignosulfonate, 2% PMDI and glucose wt.1% of the lignosulfonate weight (LO 20 P2G), compared to the particleboard made with magnesium lignosulfonate powder as adhesive (L 20). All panels have MOE values above the limit of 1600 N/mm$^2$ imposed by the standard EN 323 [60] for the P2 type panels.

The statistical analysis of the data using Minitab 18 software has shown the normal distribution of the MOR, MOE and IB values. All *p*-values resulted in the analysis were much less than 0.05, indicating that the influence of the tested lignin-based adhesives was statistically significant for all the investigated mechanical properties.

The results of bending strength (MOR) presented in Figure 3b show increased values when modified lignosulfonate by oxidation is used as adhesive, and the values increased more with the increase of cross-linkers (PMDI and glucose) content. The MOR limit imposed by standard [57] for the P2 type panels (13 N/mm$^2$) was met by panels made with adhesives composed of oxidized lignosulfonate and PMDI (2% and 3%, respectively) (panels LO 20 P2 and LO 20 P3, respectively), and also by panels with the addition of glucose and PMDI (1% and 2%) cross-linkers to oxidized lignosulfonate adhesive (panels LO 20 P1G and LO 20 P2G, respectively).

The same increased trend of MOR values was observed also for the internal bond strength (IB), as seen in Figure 3b. The limit of 0.35 N/mm$^2$ imposed by standard [57] was met for this property only for panels LO 20 P3, LO 20 P1G and LO 20 P2G. The values of the internal bond strength are for these panels 3.8 to 4.8 times higher than the value for the panel made with magnesium lignosulfonate powder as adhesive.

The results of the mechanical properties show that only the particleboards with oxidized magnesium lignosulfonate and the addition of cross-linkers, namely PMDI (3%), or PMDI (1% and 2%) and glucose (1% of the magnesium lignosulfonate content) met the requirements of the panels designed for indoor application, furniture included.

### 3.3. Microscopic Investigation and Density Profile along the Thickness

A microscopic investigation of the representative samples of the manufactured particleboards with lignin-based adhesives resulted in the images presented in Figures 4 and 5. The images in Figure 4 were obtained by magnifying 30× the central area of the sample edges. More numerous gaps between the wood particles were noticed in the case of oxidized magnesium lignosulfonate samples (panel LO 20) compared to the others. A representative example is shown in Figure 4a, where the largest gaps were measured for the structure LO20—oxidized (LIGNEX MG F) without PMDI, followed by the one with

2% PMDI, 3% PMDI and that with glucose and 2% PMDI. These gaps reflect a weaker adhesion between the particles and the adhesive.

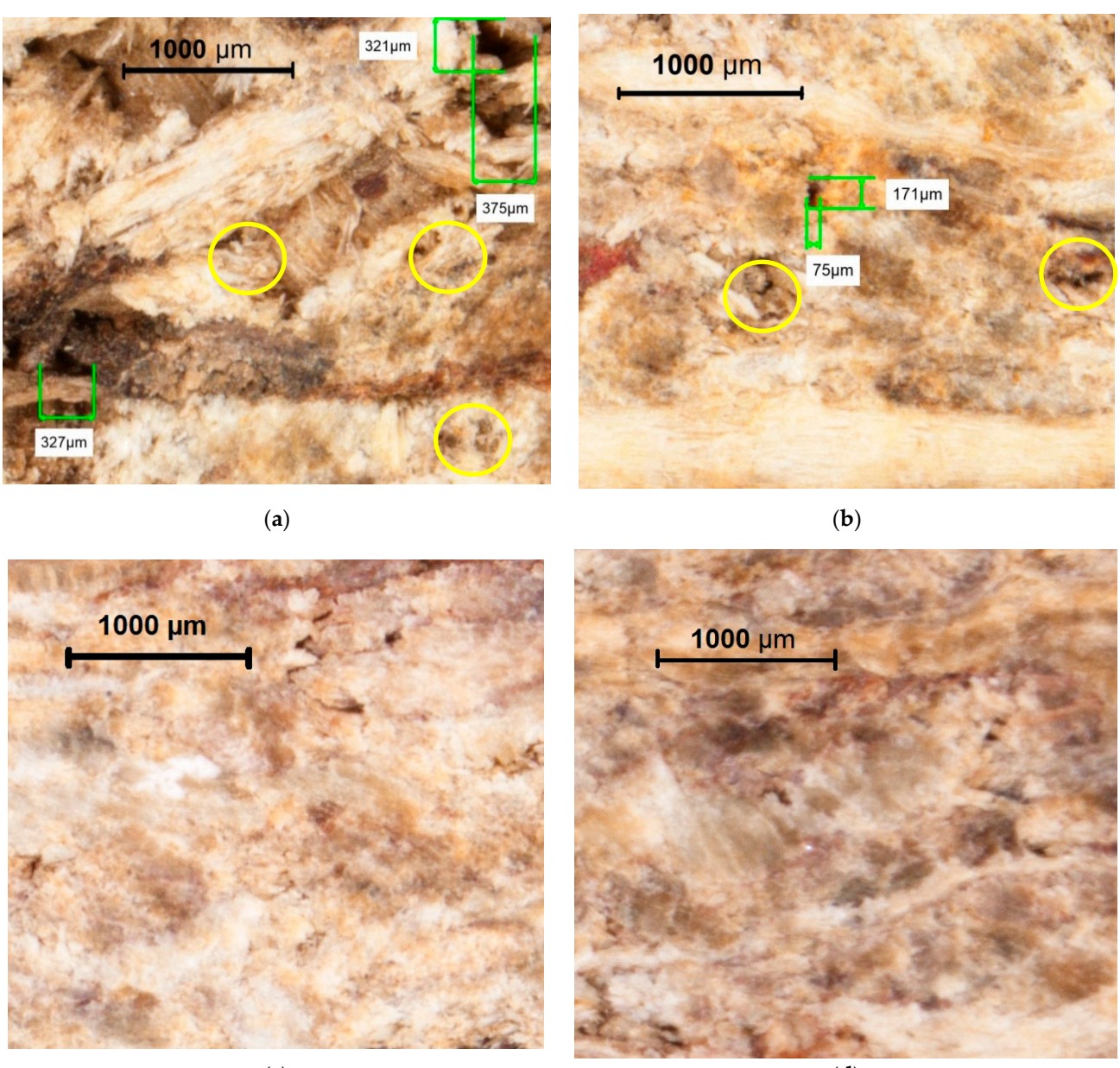

**Figure 4.** Microscopic investigation of the panels' structures, 30× magnification: (**a**) LO20; (**b**) LO 20 P2; (**c**) LO 20 P3; (**d**) LO 20 P2G. The circled areas represent the gaps between the wood particles in the structure.

Small and rare gaps were observed for the panel structure with oxidized lignosulfonate and 3% PMDI (LO 20 P3) and for the panel with oxidized lignosulfonate, 2% PMDI and glucose (LO 20 P2G), wherein crosslinked adhesive with glucose was used.

The good adhesion between the wood particles can be observed for the structures with glucose in Figure 5c,d, where 180× magnification has been used. The marked areas in Figure 5a,b highlight the adhesive agglomeration and, in consequence, the weaker interaction between particles. The more porous structures of PO 20, PO 20 P1 and PO 20 P2 panels resulted in lower values of MOR, MOE and IB.

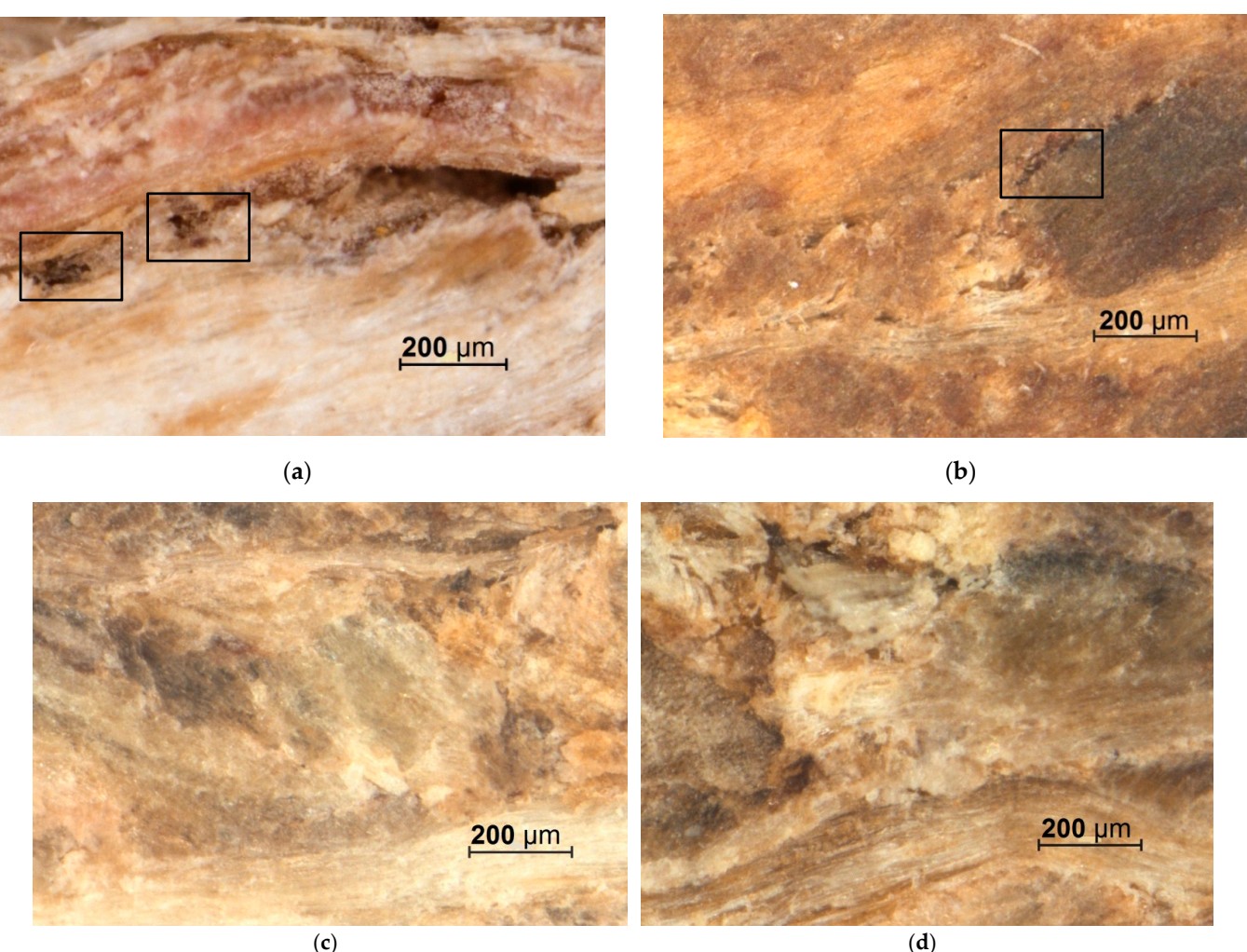

**Figure 5.** Microscopic investigation of the panels' structures, 180× magnification: (**a**) LO20; (**b**) LO 20 P2; (**c**) LO 20 P1G; (**d**) LO 20 P2G. The marked areas represent the interface between wood particles and adhesive agglomeration.

Another indicator of the mechanical performance is the density, especially the density profile along the thickness. Usually, the density of the particleboard surfaces is higher than the density of the core of the particleboard. The same conclusion was found by other authors [67]. The higher surface density and lower core density (Figure 6a) resulted in better bending strength (MOR), in general observed for oxidized magnesium lignosulfonate panels. The profile with lower surface densities and higher core densities has been obtained for the panels wherein both PMDI and glucose were used as cross-linkers (PO 20 P1G and P2G), which led to the improvement of the internal bond strength (IB) [67] (Figure 6b).

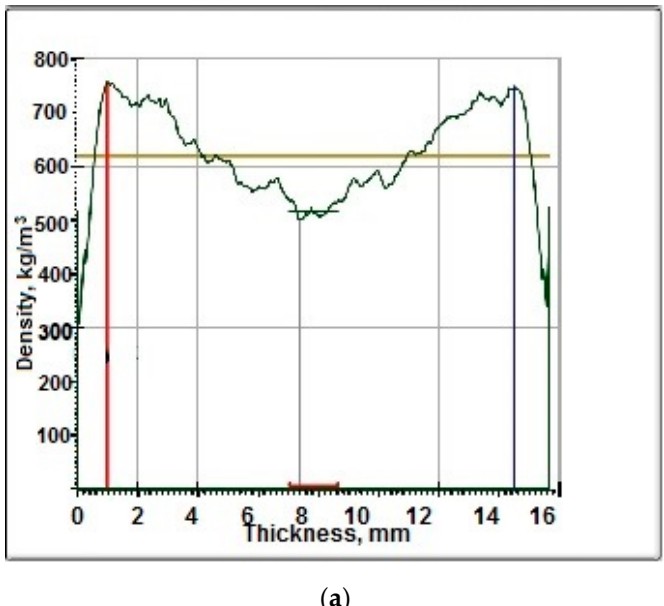
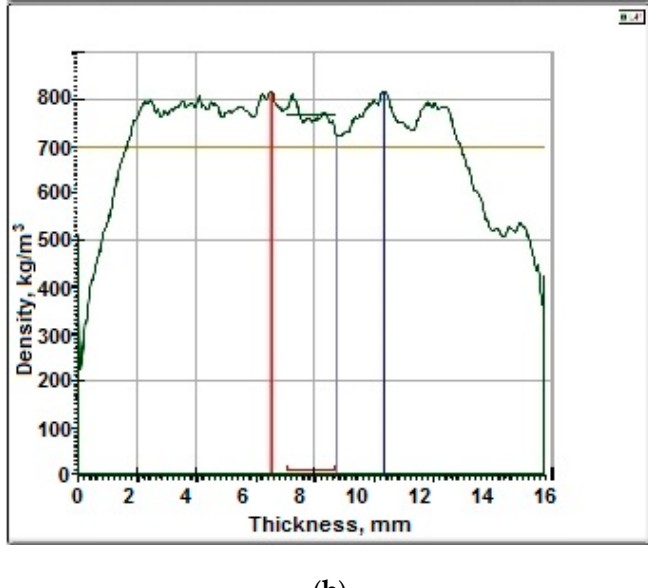

(**a**)                                                                                          (**b**)

**Figure 6.** Panel density profiles along the panel thickness leading to very different board properties; (**a**) panel with oxidized lignin (LO20); (**b**) panel with oxidized lignin, 2% PMDI and glucose (PO 20 P2G).

### 3.4. Free Formaldehyde Emission

The experimental values of the formaldehyde emission are presented in Table 6.

**Table 6.** Formaldehyde emission, in mg/ $m^2 \cdot h$.

| Type of the Receipt | Values |
|---|---|
| LP 20 (LIGNEX MG F powder) | 0.789 |
| LO 20 (oxidized LIGNEX MG F) | 0.616 |
| LO 20 P1 (oxidized LIGNEX MG F + 1% PMDI) | 0.553 |
| LO 20 P2 (oxidized LIGNEX MG F + 2% PMDI) | 0.477 |
| LO 20 P3 (oxidized LIGNEX MG F + 3% PMDI) | 0.401 |
| LO 20 P1G (oxidized LIGNEX MG F + 1% PMDI + glucose) | 0.386 |
| LO 20 P2G (oxidized LIGNEX MG F + 2% PMDI + glucose) | 0.347 |

It was observed that the panels with PMDI and glucose cross-linkers (LO 20 P1G and P2G) had the lowest formaldehyde emission compared to those with pure lignin (L 20), with about 40%. Increasing the amount of PMDI with 2%, led to a decrease of formaldehyde emission with about 28% (LO 20P1 and LO 20 P3 compared to LO 20P1). Because of the PMDI's superior moisture tolerance, it was not necessary to dry the wood particles to levels lower than 4%; the moisture content of wood particles ranged in the limits of 10% to 15% which is ideal for bonding [68]. High moisture content facilitates the hydrolysis speed of wood components and the transport of the formaldehyde out of the particleboard panel [69]. As shown in Figure 7, the high degree of correlation ($R^2$ = 0.97) between the quantity of formaldehyde released and the PMDI ratio was obtained. The decrease of formaldehyde emission with increase in PMDI level could be explained by the faster curing of the adhesive film formed by PMDI on the particle surfaces, which may have prevented VOCs and the release of formaldehyde from the particle surfaces [69]. These could be also deduced from the microscopic images (Figure 5c,d), which showed the good interface of particles. Several researchers [70–72] have reported that bio-based formaldehyde scavengers with hydroxyl groups contribute both to the high internal cohesion of boards and the low formaldehyde emission of particleboards. Based on this theory, the D-glucose, which has hydroxyl groups in its composition, may act as formaldehyde scavenger [73], thus lowering the formaldehyde emission, as can be seen for LO 20 P1G and LO 20 P2G panels (Table 5

and Figure 7). Formaldehyde emission values are much lower than the standardized limit specified for E1 class (3.5 mg/m$^2$·h) [61]. It can be noticed that the formaldehyde values obtained are close to those of some natural wood species (Douglas fir and oak: 0.397 mg/m$^2$·h and 0.43 mg/m$^2$·h, respectively) [74].

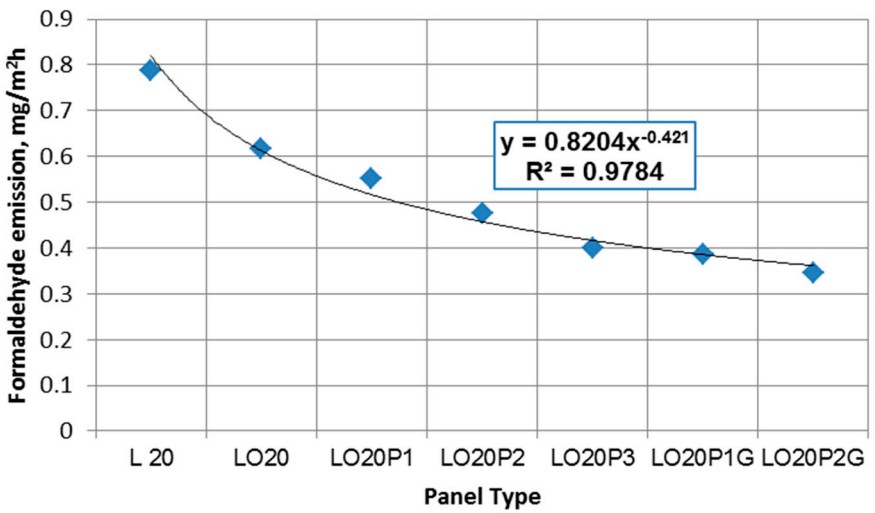

**Figure 7.** Power curve fitting of formaldehyde emission.

## 4. Conclusions

The research presented in this paper shows that magnesium lignosulfonate has a great potential to be used as an adhesive in particleboard manufacturing when cross-linkers such as PMDI and glucose are added in the adhesive recipes. The process of mild oxidation of lignosulfonate with hydrogen peroxide caused only minor changes in its chemical structure and its bonding properties. Particleboards glued only with oxidized lignosulfonate (20% reported in oven-dried wood) without cross-linkers did not meet the required standard performance. An addition of cross-linkers, such as PMDI, of 3% (reported in the weight of dried wood particles), or mixed PMDI (1% and 2%, respectively, of the weight of the dried wood particles) and glucose (1% of the lignosulfonate content), improved the bonding properties enough to meet the requirements of the panels P2 designed for indoor use (furniture included), according to EN 312: 2004. FTIR analysis confirmed cross-linking reactions involving PMDI, the oxidized magnesium lignosulfonate and glucose.

Formaldehyde emission released from the experimental panels ranged between 0.347 mg/m$^2$·h and 0.789 mg/m$^2$·h, the lower limits corresponding to values of formaldehyde generated by several natural wood species, as other researchers also stated in their study [74]. The experimental results proved that cross-linkers, such as PMDI and glucose contribute not only to a higher internal cohesion of boards, but they may act as formaldehyde scavengers.

The paper promotes a solution for the use of lignin in eco-adhesives for wood-based panels, currently limited to indoor applications. Further tests must be done in order to extend their uses for outdoor applications. This study aligns with the new regulations concerning formaldehyde emission, which requires reaching E0 class in the future.

**Author Contributions:** Conceptualization, C.C. and G.B.P.; methodology, C.C., G.B.P., O.Z., M.C.T. and A.L.; software, C.C., M.C.T. and G.B.P.; validation, C.C., G.B.P., O.Z., M.C.T. and A.L.; formal analysis, C.C. and O.Z.; investigation, G.B.P., M.C.T. and A.L.; resources, G.B.P.; data curation, C.C.; writing—original draft preparation, C.C. and O.Z.; writing—review and editing, C.C., M.C.T. and O.Z.; visualization, O.Z. and M.C.T.; supervision, C.C.; project administration, G.B.P.; funding acquisition, G.B.P. All authors have read and agreed to the published version of the manuscript.

**Funding:** This research received no external funding.

**Acknowledgments:** We hereby acknowledge the structural funds project PRO-DD (POS-CCE, O.2.2.1., ID 123, SMIS 2637, No. 11/2009) for providing the infrastructure used in this work and Contract No. 7/9.01.2014.

**Conflicts of Interest:** The authors declare no conflict of interest.

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
