# Peer review of "Mechanical Properties and Formaldehyde Release of Particleboard Made with Lignin-Based Adhesives"

_applsci, doi:10.3390/app11188720_

Round 1
Reviewer 1 Report
I had the opportunity to do the review of the article "Mechanical Properties and Formaldehyde Release of Particleboard Made with Lignin-based Adhesives". The article is well written. Needs some improvements before publishing.
Suggestions:
The abstract: Please discuss more results of your research, add a discussion from FTIR, microscopic investigation, and density profile.
The Introduction:
Line 24: not only UF but also PF and MF (together represents 95% of total adhesives used in wood-based panel industry) due to their numerous advantages (low price, short press times, low press temperature, excellent adhesion, etc.).
Line 25: Please mention "various methods" (like adding scavengers, surface treatment of wood-based panels, use of eco-friendly adhesives, etc.)
Line 38: Resins or adhesives, please be consistent throughout the whole manuscript.
Line 39: Please discuss also well-known work:
doi.org/10.1016/j.ijadhadh.2019.102408
where authors found that up to 50% wt of the phenol can be replaced by kraft lignin or work:
10.1002/app.20374
where 50wt % with Eucalyptus bark lignin in the polycondensation process.
Line 34-43: Please mention also that unmodified lignin has not only low reactivity, also a longer pressing time and higher pressing temperature needed for the resin produced from it.
Line 34-49: Please mention that there are 6 most used modifications to enhance the reactivity of lignin i.e. demethylation, methylolation/hydroxymethylation, phenolation/phenolysis, reduction, and hydrolysis.
Lines 50-68: Please discuss also use of other LS (i.e. ammonium, calcium, magnesium), please check here:
doi.org/10.3390/polym13040639
Yuan et al. (2014), BioResources 9(1), 836-848
doi.org/10.1016/j.indcrop.2015.09.075
or addition of LS (i.e. ammonium, magnesium, sodium) to UF adhesives for lowering formaldehyde emission:
doi.org/10.3390/ma14174875
doi.org/10.3390/polym13020220
Lines 63-68: Please discuss also other crosslinkers.
Line 106: adhesive reactivity is 3 min 11s, in Table 3 it is 3 min 15s.
Line 123: Please add the type of mixer used.
Line 124: Please add data for pre-press.
Lines 180-195: you are talking about free formaldehyde. Also, I suggest using emission instead of release.
I suggest adding part 2.7 about statistical methods used.
Figure 2: The figure is hard to read.
Table 5: Are the differences statistically significant?
Line 331: Free formaldehyde...
Lines 337-359: Please discuss the statistical significance of the results.
Results and discussion: Please add more discussion with other research in this area.
The Conclusions: Please add implications for practice and also limitations of your research.
Author Response
Response to Reviewer 1 Comments
Thank to reviewer for all remarks and suggestions made, which helped us to improve our article. The changes were made in the revised paper by red fonts. The words and phrases deleted and yellow highlighted have to be removed from the text.
- Reviewer: The abstract: Please discuss more results of your research, add a discussion from FTIR, microscopic investigation, and density profile.
Authors: The abstract was supplemented with other results related to FTIR, microscopic and density profile analyses, as follows:
The spectroscopic analysis has revealed structural changes brought by oxidation which may indicate depolymerization by splitting of C-O-C bonds leading to carbonyl groups. Mechanical properties were improved, and the highest average values were recorded for panels having as adhesives oxidized lignin with cross-linkers, as follow: 15 N/mm2 (MOR), 3320 N/mm2 (MOE) and 0.48N/mm2 (IB). Density profile presented higher values for faces in case of oxidized lignin panels. Changes were observed for oxidized lignin with cross-linkers panels where the core had higher values.
- Reviewer: The Introduction: Line 24: not only UF but also PF and MF (together represents 95% of total adhesives used in wood-based panel industry) due to their numerous advantages (low price, short press times, low press temperature, excellent adhesion, etc.).
Authors: The suggestion regarding to other adhesives used namely phenol-formaldehyde (PF) and melamine-formaldehyde (MF) in the wood-based composite board industry has been introduced in the paper.
Formaldehyde issues are related to urea-formaldehyde resin (UF) as dominating bonding adhesive used for the production of wood-based panels (Pizzi and Mittal 2003, Mantanis et al. 2017). Melamine-formaldehyde (MF), melamine-urea-formaldehyde (MUF) and phenol-formaldehyde (PF) adhesives are less dangerous in terms of their subsequent formaldehyde emission but they were not convinced industrial producers, due to their higher price or lower reactivity (Despres et al. 2010; Costa et al. 2013).
- Reviewer: Line 25: Please mention "various methods" (like adding scavengers, surface treatment of wood-based panels, use of eco-friendly adhesives, etc.)
Authors: The completion suggested by the reviewer was introduced in the paper in the introductory area, as follows:
”…, the use of formaldehyde scavengers [Boran et al. 2011, Boran et al. 2011, Costa et al. 2013, Neimsuwan et al. 2017, Antov et al. 2020], the surface treatment of wood composites [Park et al, 2016, Chen et al. 2020],…”
- Reviewer: Line 38: Resins or adhesives, please be consistent throughout the whole manuscript.
Authors: The word "resin" was replaced with "adhesives" throughout the paper.
- Reviewer: Line 39: Please discuss also well-known work: doi.org/10.1016/j.ijadhadh.2019.102408 where authors found that up to 50% wt of the phenol can be replaced by kraft lignin or work: 10.1002/app.20374 where 50wt % with Eucalyptus bark lignin in the polycondensation process.
Authors: The two papers recommended by the reviewer were analysed and presented in the paper, as follows:
Ang et al [24] referred to lignin-based copolymer adhesives for the production of wood based composites and found that up to 50% wt. of the phenol can be replaced by kraft lignin. Khan et al [25] found that up to 50% of phenol in phenol formaldehyde adhesive can be substituted by bark lignin with improved adhesive properties.
- Reviewer: Line 34-43: Please mention also that unmodified lignin has not only low reactivity, also a longer pressing time and higher pressing temperature needed for the resin produced from it.
Authors: The mention of the reviewer was introduced in the paper in the indicated line.
- Reviewer: Line 34-49: Please mention that there are 6 most used modifications to enhance the reactivity of lignin i.e., demethylation, methylenation /hydromethylation, phenylation/ phenolics, reduction, and hydrolysis.
Authors: The phrase was modified, as follows:
The most used methods to fulfill this objective are [18,19] methylolation (hydroxymethylation), phenolation, demethylation, reduction, hydrolysis and oxidation [20-22, Hu 2011 ].
- Reviewer: Lines 50-68: Please discuss also use of other LS (i.e. ammonium, calcium, magnesium), please check here: doi.org/10.3390/polym13040639
Yuan et al. (2014), BioResources 9(1), 836-848 doi.org/10.1016/j.indcrop.2015.09.075 or addition of LS (i.e. ammonium, magnesium,sodium) to UF adhesives for lowering formaldehyde emission:
doi.org/10.3390/ma14174875; doi.org/10.3390/polym13020220
Authors: A phrase was introduced in the text:
Calcium lignosulfonate [26] magnesium and sodium lignosulfonates [27], ammonium lignosulfonate [28, 29] or modified ammonium lignosulfonate [30] were used as adhesives for lowering formaldehyde emission and improve the mechanical properties of the panels..
- Reviewer: Lines 63-68: Please discuss also other crosslinkers.
Authors: Other cross-linkers have been mentioned in the text:.
Other cross-linkers are mentioned in the literature: alternative aldehydes (glyoxal, glutaraldehyde), polyacids (citric acid, maleic anhydride) (Solt et al).
- Reviewer: Line 106: adhesive reactivity is 3 min 11s, in Table 3 it is 3 min 15s.
Authors: The modification was done to 15s.
- Reviewer: Line 123: Please add the type of mixer used.
Authors: Data about the used mixer was added.
- Reviewer: Line 124: Please add data for pre-press.
Authors: The data were added.
- Reviewer: Lines 180-195: you are talking about free formaldehyde. Also, I suggest using emission instead of release.
Authors: In this paragraph we replaced word “release” with “emission”.
- Reviewer: I suggest adding part 2.7 about statistical methods used.
Authors: Subchapter 2.7 on the statistical interpretation of results has been introduced, as follows:
2.7. Statistical methods
Minitab 18 statistical software (Coventry, United Kingdom) was used for the interpretation of the differences between tested lignin-based adhesives, related to mechanical properties. A confidence interval of 95% was used for the statistical analysis, with the acceptance of an alpha type error of 0.05. The Anderson-Darling and p-value parameters analyzed the normality of the distribution of the results and whether it has statistically significant differences.
- Reviewer: Figure 2: The figure is hard to read.
Authors: The figure was generated by FTIR analysis. It can not be changed.
- Reviewer: Table 5: Are the differences statistically significant?
Authors: Statistical analysis was performed with the Minitab 18 program, comparing the p-value value with the alpha error, taking into account the confidence interval, too.
- Reviewer: Line 331: Free formaldehyde...
Authors: In this line we added the word “free” before formaldehyde and replaced word “release” with “emission”.
- Reviewer: Lines 337-359: Please discuss the statistical significance of the results.
Authors: The statistical results were also added in the text:
The statistical analysis of the data using Minitab 18 software has shown the normal distribution of the MOR, MOE and IB values. All p-values resulted in the analysis were much less than 0.05, indicating that the influence of the tested lignin-based adhesives was statistically significant for all the investigated mechanical properties.
- Reviewer: Results and discussion: Please add more discussion with other research in this area.
Authors: Several discussions were introduced in the research and discussion chapter. In addition, in some places the discussion was highlighted, for a better visibility.
- Reviewer: The Conclusions: Please add implications for practice and also limitations of your research.
Authors: A new paragraph was introduced in the conclusions chapter:
The paper promotes a solution for the capitalization of lignin into eco-adhesives for wood based panels, being limited to indoor applications. Further tests must be done in order to extend their uses for outdoor applications. The study aligns to the new regulations concerning the formaldehyde emission, which limits have to reach E0 class in the future.
Thank you for your recommendations,
The authors

Reviewer 2 Report
The manuscript is focused on an innovative research topic, namely investigation and evaluation of the mechanical properties and free formaldehyde emission of wood-based composites (particleboards), manufactured with formaldehyde-free, lignin-based wood adhesives. In this respect, the presented manuscript is within the scope and aims of the Applied Sciences Journal.
The manuscript can be accepted for publication in the Applied Sciences Journal after minor revision. Please, see below my comments on your work:
In general, the abstract (lines 9-18) and the keywords (lines 19-20) correspond to the title, aims and objectives of the manuscript. However, the abstract is a bit short, and does not reflect all findings of the study. I’d recommend to the authors to extend the abstract of the manuscript with specific results of their study.
In addition, I’d propose to add “bio-based adhesives” and “wood-based composites” in the keywords.
In line 12, please replace the word “recipes” with “adhesive formulations”.
In line 23, the statement about formaldehyde emission in generally true, but needs further explanation, i.e. the negative effects of free formaldehyde emitted from wood-based panels on human health, including cancer. Namely this was the reason for setting the new restrictions related to formaldehyde emission from wood-based composites in Europe, USA, and Japan. Please add some explanation.
In line 24, the authors are right that currently, the UF resins are the most predominant type of formaldehyde-based wood adhesives worldwide due to their numerous advantages. In addition, I’d recommend to add also phenol-formaldehyde (PF), melamine-formaldehyde resins (MF), and melamine-urea-formaldehyde (MUF) resins, since they are also among the most used adhesives for wood panels.
In lines 32-34, in addition to the described methods for decreasing the formaldehyde emission, I’d recommend to the authors to include also the use of formaldehyde scavengers and surface treatment of wood composites.
In line 42, please give the full name of the resin, i.e. phenol-formaldehyde resin, then the common abbreviation PF.
In lines 44-45, please also add methylolation and phenolation as other chemical modifications of lignin aimed at increasing its reactivity to formaldehyde.
In addition to the previous studies, discussed in the Introduction, presenting the use of lignosulfonates as adhesives in wood-based panels, please add some information and references about the drawbacks, limiting their wider industrial use – longer pressing times, significantly deteriorated dimensional stability, etc.
A recent research about wood-based panels bonded with UF resin and ammonium lignosulfonate reported successful attempt to produce panels with acceptable physical and mechanical properties and very low free formaldehyde emission, fabricated at very short press time, comparable with industrial conditions. Please refer to the following: https://doi.org/10.3390/polym13162775
In line 61, please add information about the super E0 emission class requirements, i.e. ≤4 mg/100 g, measured according to the Perforator method (EN ISO 12460-5).
Another recent study reported production of eco-friendly particleboards bonded with lignosulfonates, UF resin, and pMDI as a crosslinker. Please refer to the following: https://doi.org/10.3390/ma14174875
Overall, the Introduction part is well written and informative, but can be further elaborated based on the comments provided. The inclusion of additional references on the topic is also recommended.
In line 79, please add relevant information about the vibrating sieves (company producer, city, country).
In line 96, Table 2. Characteristics of the Magnesium Lignosulfonate, there is included “Magnesium and calcium (total)”. Please explain the presence of calcium.
In line 108, please add some information and characteristics about the pMDI used in the research.
In Table 3, the value of adhesive reactivity is 3 min 15 s, while in line 106 it is 3 min and 11 s, please revise accordingly.
In line 123, please add some information about the equipment used for mechanical mixing.
In line 125, please add some information about the laboratory press used (company producer, city, country).
In line 131-132, the information about the percentage of lignosulfonate has already been given in lines 117-118, please revise.
In line 145, please add information about the laboratory oven used.
In line 236, Figure 2 is not of a very good quality, please replace if possible.
In lines 261-296, please add discussion about the mechanical properties of the laboratory-produced particleboards with previously published studies in the field.
In general, the Results and Discussion is well-written, clear and informative. Additional discussion with previous studies is recommended.
In general, the Conclusions (lines 360-378) are consistent with the results and reflect the main findings of the study.
The references cited are appropriate and correspond to the topic of the manuscript. The inclusion of additional references in all sections of the manuscript is highly recommended. This will significantly increase the value of the presented article.
The standards used (references 33-39) are not properly cited. The British Standards should be listed as “BS EN…”. I’d recommend to refer to the EN standards.
Best regards!
Author Response
Response to Reviewer 2 Comments
Thank to reviewer for all remarks and suggestions made, which helped us to improve our article. The changes were made in the revised paper by red fonts. The words and phrases deleted and yellow highlighted have to be removed from the text.
Point 1: However, the abstract is a bit short, and does not reflect all findings of the study. I’d recommend to the authors to extend the abstract of the manuscript with specific results of their study. I’d propose to add “bio-based adhesives” and “wood-based composites” in the keywords.
Response 1: We added in the keywords: “bio-based adhesives” and “wood-based composites”. We add the following results in the abstract: The spectroscopic analysis has revealed structural changes brought by oxidation which may indicate depolymerization by splitting of C-O-C bonds leading to carbonyl groups. Mechanical properties were improved, and the highest average values were recorded for panels having as adhesives oxidized lignin with cross-linkers, as follow: 15 N/mm2 (MOR), 3320 N/mm2 (MOE) and 0.48N/mm2 (IB). Density profile presented higher values for faces in case of oxidized lignin panels. Changes were observed for oxidized lignin with cross-linkers panels where the core had higher values.
Point 2: In line 12, please replace the word “recipes” with “adhesive formulations”
Response 2: We replaced “recipes” with “adhesive formulations” .
Point 3: In line 23, the statement about formaldehyde emission in generally true, but needs further explanation, i.e. the negative effects of free formaldehyde emitted from wood-based panels on human health, including cancer. Namely this was the reason for setting the new restrictions related to formaldehyde emission from wood-based composites in Europe, USA, and Japan. Please add some explanation.
In line 24, the authors are right that currently, the UF resins are the most predominant type of formaldehyde-based wood adhesives worldwide due to their numerous advantages. In addition, I’d recommend to add also phenol-formaldehyde (PF), melamine-formaldehyde resins (MF), and melamine-urea-formaldehyde (MUF) resins, since they are also among the most used adhesives for wood panels.
Response 3:
Lines 23-24 Thank to reviewer for recommendations.
We introduced the following paragraph about formaldehyde and others adhesives:
Formaldehyde issues are related to urea-formaldehyde resin (UF) as dominating bonding adhesive used for the production of wood-based panels (Pizzi and Mittal 2003, Mantanis et al. 2017). Melamine-formaldehyde (MF), melamine-urea-formaldehyde (MUF) and phenol-formaldehyde (PF) adhesives are less dangerous in terms of their subsequent formaldehyde emission but thet were not convinced industrial producers due to their higher price or lower reactivity (Despres et al. 2010; Costa et al. 2013).
Exposure to formaldehyde is possible during the adhesives mixing, mat forming and hot-pressing operations. Formaldehyde exposure can be harmful to human health. At low levels, formaldehyde can cause eye, nose and throat irritation, and skin rashes, shortness of breath, wheezing and changes in lung function at higher levels exposure (https://www.atsdr.cdc.gov/formaldehyde/. The International Agency for Research on Cancer (IARC) has reclassified the formaldehyde into the Group 1-"carcinogenic to humans" (https://monographs.iarc.who.int/wp-content/uploads/2018/06/mono88.pdf
Environmental Protection Agency (EPA) considered also formaldehyde to be a probable human carcinogen (cancer-causing agent) and has ranked it in EPA's Group B1 (U.S. Environmental Protection Agency. Integrated Risk Information System (IRIS) on Formaldehyde. National Center for Environmental Assessment, Office of Research and Development, Washington, DC. 1999. din https://www.epa.gov/sites/default/files/2016-09/documents/formaldehyde.pdf. In Europe, the European Chemicals Agency (ECHA) reclassified formaldehyde in category 1B “presumed human carcinogen” and “germ cell mutagen category 2 (acute toxicity)” in 2012 (https://www.formacare.eu/about-formaldehyde/regulatory-status/clp/) With growing interest in the indoor air quality, efforts have been made to reduce exposure limits to formaldehyde both in the workplaces and in the living spaces. There are legislations in the European Union, USA, China, and Japan limiting the levels of formaldehyde emission (FE) from wood and wood-based products.
Point 4: In lines 32-34, in addition to the described methods for decreasing the formaldehyde emission, I’d recommend to the authors to include also the use of formaldehyde scavengers and surface treatment of wood composites.
Response 4: As recommended we added in Lines 32-34 the formaldehyde scavengers and surface treatment of wood composites, as follow:
… the use of formaldehyde scavengers [Boran et al. 2011, Boran et al. 2011, Costa et al. 2013, Neimsuwan et al. 2017, Antov et al. 2020], the surface treatment of wood composites [Park et al, 2016, Chen et al. 2020], ….
Point 5: In line 42, please give the full name of the resin, i.e. phenol-formaldehyde resin, then the common abbreviation PF.
In lines 44-45, please also add methylolation and phenolation as other chemical modifications of lignin aimed at increasing its reactivity to formaldehyde.
Response 5: We have modified according to the recommendations
The most used methods to fulfill this objective are [18,19] methylolation (hydroxymethylation), phenolation, demethylation, reduction, hydrolysis and oxidation [20-22, Hu 2011 ].
Point 6. In addition to the previous studies, discussed in the Introduction, presenting the use of lignosulfonates as adhesives in wood-based panels, please add some information and references about the drawbacks, limiting their wider industrial use – longer pressing times, significantly deteriorated dimensional stability, etc.
Response 6: We added some comments about lignosulfonates as suggested: The wider industrial application of lignosulphonates in the composition of wood adhesives is limited by the increased hydrophilicity of finished wood-based panels, longer press times and the availability of crosslinkers [Hemmilä et al.2017, Antov et al. 2020].
Point 7. A recent research about wood-based panels bonded with UF resin and ammonium lignosulfonate reported successful attempt to produce panels with acceptable physical and mechanical properties and very low free formaldehyde emission, fabricated at very short press time, comparable with industrial conditions.
Response 7: In line 63 was added the following comment as suggested.
[Antov et al. 2021] was added as reference in the following text:
Calcium lignosulfonate [26] magnesium and sodium lignosulfonates [27], ammonium lignosulfonate [28, 29, Antov: https://doi.org/10.3390/polym13162775] or modified ammonium lignosulfonate [30] were used as adhesives for lowering formaldehyde emission and improve the mechanical properties of the panels.
Point 8. In line 61, please add information about the super E0 emission class requirements, i.e. ≤4 mg/100 g, measured according to the Perforator method (EN ISO 12460-5).
Response 8: The following phrase was added: Pressure from environmental and health regulators, causes formaldehyde emission requirements to become more stringent. Thus, super-E0 (≤1.5 mg/100 g acc. to Perforator method EN ISO 12460-5. or ≤ 0.3 mg/L acc. to Desiccator method JIS A 1460) and E0 (≤2.5 mg/100 g, or ≤ 0.5 mg/L) classes of panels are promoted [https://www.ntl-chemicals.com/wp-content/uploads/2019/02/MDF_Yearbook_2017_2018_Article_NTL_Chemical_Print-1.pdf]
Point 9: Another recent study reported production of ecofriendly particleboards bonded with lignosulfonates, UF resin, and pMDI as a crosslinker. Please refer to the following: https://doi.org/10.3390/ma14174875
Response 9: It was added the reference indicated ….PMDI is a promising cross-linker that works for most bio-based adhesives [29,30… Bektha]
Bekhta, P.; Noshchenko, G.; Réh, R.; Kristak, L.; Sedliačik, J.; Antov, P.; Mirski, R.; Savov, V. Properties of Eco-Friendly Particleboards Bonded with Lignosulfonate-Urea-Formaldehyde Adhesives and pMDI as a Crosslinker. Materials 2021, 14, 4875. https://doi.org/10.3390/ma14174875
Point 10: In line 79, please add relevant information about the vibrating sieves (company producer, city, country).
Response 10: We fulfilled the data according to suggestion of the reviewer – …vibrating sieves (VEB Metallweberei Neustadt Orla, Germany).
Point 11: In line 108, please add some information and characteristics about the pMDI used in the research
In line 123, please add some information about the equipment used for mechanical mixing.
In line 125, please add some information about the laboratory press used (company producer, city, country).
In line 145, please add information about the laboratory oven used.
Response 11: We added the information required
Point 12: In Table 3, the value of adhesive reactivity is 3 min 15 s, while in line 106 it is 3 min and 11 s, please revise accordingly.
In line 131-132, the information about the percentage of lignosulfonate has already been given in lines 117-118, please revise.
Response 12: We revised the data and deleted the phrase from line 131-132.
Point 13: In line 236, Figure 2 is not of a very good quality.
Response 13: The figure was generated by FTIR analysis. It can not be changed.
Point 14: In lines 261-296, please add discussion about the mechanical properties of the laboratory-produced particleboards with previously published studies in the field.
Response 14: The following data were added:
The values of MOR and MOE are comparable with results obtained for composite panels produced from waste fibres bonded with magnesium lignosulfonate [Antov et al 2020,
Antov, P., Mantanis, G.I. and Savov, V. Development of Wood Composites from Recycled
Fibres Bonded with Magnesium Lignosulfonate. Forests 2020, 11, 613; doi:10.3390/f11060613 - http://mantanis.users.uth.gr/R2020-04.pdf
Point 15: The standards used (references 33-39) are not properly cited. The British Standards should be listed as “BS EN…”. I’d recommend to refer to the EN standards
Response 15: The modification was done to all European standards.
Thank you for your recommendations,
The authors

Round 2
Reviewer 1 Report
The manuscript was improved.